# A single spin in hexagonal boron nitride for vectorial quantum magnetometry

Carmem M. Gilardoni [1,2,8] ✉, Simone Eizagirre Barker[1,8], Catherine L. Curtin[1], Stephanie A. Fraser [1], Oliver. F. J. Powell [1,3], Dillon K. Lewis[1], Xiaoxi Deng[1], Andrew J. Ramsay [3], Sonachand Adhikari[4], Chi Li [5,6], Igor Aharonovich[5,6], Hark Hoe Tan [4], Mete Atatüre [1] & Hannah L. Stern [7] ✉

Quantum sensing based on solid-state spin defects provides a uniquely versatile platform for nanoscale magnetometry under diverse environmental conditions. Operation of most sensors used to-date is based on projective measurement along a single axis combined with computational extrapolation. Here, we show that an individually addressable carbon-related spin defect in hexagonal boron nitride is a multi-axis nanoscale sensor with large dynamic range. For this spin-1 system, we demonstrate how its spin-dependent photodynamics give rise to three optically detected spin resonances that show up to 90% contrast and are not quenched under off-axis magnetic field exceeding 100 mT, enabling $\mu T/Hz^{-1/2}$ sensitivity. Finally, we show how this system can be used to unambiguously determine the three components of a target magnetic field via the use of two bias fields. Alongside these features, the room-temperature operation and the nanometer-scale proximity enabled by the van der Waals host material further consolidate this system as a promising quantum sensing platform.

Spin defects in solids can be used as quantum sensors to study phenomena across condensed matter, geological, and biological systems[1–3]. When reduced to the single spin level, optically addressable high-spin ($S > 1/2$) defects can provide quantitative field, temperature, and pressure sensors with nanoscale spatial resolution, in a technique described as nanoscale quantum sensing[4]. The rapid development of quantum sensors has been driven largely by the nitrogen-vacancy (NV) centre in diamond[5–12], as well as defects in silicon carbide[13–15]. For DC sensing, pioneering work with the NV centre has demonstrated mapping of static fields formed by spin order and current flow in materials[16–23], including in atomically thin semiconductors[24,25]. Much of

this work has provided key fundamental insight into the nature of magnetisation in these materials[26,27].

A challenge for nanoscale magnetometry is access to sensors that can detect all three components of a magnetic field vector[28,29]. The most established systems, the NV centre and superconducting quantum interference device sensors, are single-axis sensors where computational extrapolation is required to map the full vector field. This can make retrieval of magnetisation information more susceptible to noise and introduce error[28–30]. A second challenge that faces the NV centre specifically is that sensor operation is limited to experimental conditions where off-axis magnetic field does not exceed ~10 mT[17,31].

[1]Cavendish Laboratory, University of Cambridge, JJ Thomson Avenue, Cambridge CB3 0HE, UK. [2]Centro Brasileiro de Pesquisas Físicas, Rua Dr. Xavier Sigaud, 150, Urca, Rio de Janeiro 22290-180 RJ, Brazil. [3]Hitachi Cambridge Laboratory, Hitachi Europe Ltd., JJ Thomson Avenue, Cambridge CB3 0HE, UK. [4]ARC Centre of Excellence for Transformative Meta-Optical Systems, Department of Electronic Materials Engineering, Research School of Physics, The Australian National University, Canberra, ACT, Australia. [5]ARC Centre of Excellence for Transformative Meta-Optical Systems, Faculty of Science, University of Technology Sydney, Ultimo, NSW, Australia. [6]School of Mathematical and Physical Sciences, Faculty of Science, University of Technology Sydney, Ultimo, NSW, Australia. [7]Department of Materials, University of Oxford, Parks Road, Oxford OX1 3PH, UK. [8]These authors contributed equally: Carmem M. Gilardoni, Simone Eizagirre Barker. ✉e-mail: gilardonicm@cbpf.br; hannah.stern@materials.ox.ac.uk

This is because strong off-axis magnetic field can lead to spin mixing that degrades the optical initialisation process of the NV centre[17]. Increasing the target-defect distance is one way to reduce this susceptibility to transverse fields, however this limits the spatial resolution. A high-spin defect that can overcome the limitations of dynamic range while providing full vectorial sensitivity and <10 nm spatial resolution with operation over a broad temperature range would dramatically increase the throughput and the scope of nanoscale quantum sensing.

Spin defects in two-dimensional materials are a new platform for nanoscale quantum sensing, where the atomic thickness and layered nature of the host material may enable higher spatial resolution and provide new opportunities for integration into hybrid devices[32,33]. Hexagonal boron nitride (hBN) is emerging as a material that offers a range of optically active spin defects with different attributes for quantum sensing. Wide-field quantum microscopy with the $S=1$ optically addressable boron vacancy ($V_B^-$) spin defect ensembles in hexagonal boron nitride (hBN) shows the versatility of the hBN host material regarding integration with 2D heterostructures[34–36]. These reports present mapping of magnetic domains, temperature, and charge currents in layered ferromagnets, albeit with diffraction-limited spatial resolution due to the use of defect ensembles. More recent reports have revealed a new class of defects with a $S$-1/2-like signature that can be used for magnetic imaging[37,38]. These defects, although present at the single-defect level, do not possess a strong quantisation axis and therefore cannot provide information on the orientation of the applied magnetic field.

In this article, we reveal that the $S=1$ carbon-related spin defect in hBN[37,39] is an attractive system for nanoscale quantum sensing, displaying vectorial sensing capability, broad magnetic field dynamic range, competitive DC sensitivity, and potential for unprecedented spatial resolution. We show that, due to its advantageous excited-state dynamics, this spin defect displays multiple ground-state optically detected magnetic resonances with contrast that can exceed 90% (see

Supplementary Note 1) and a signal that persists at arbitrarily orientated magnetic field beyond 100 mT. We combine photon-emission correlation spectroscopy and pulsed ODMR experiments with microscopic modelling of the optical cycle to explain the kinetic origin of the high ODMR contrast and large dynamic range for this quantum sensor. Finally, we show that full vectorial mapping of a target field is accessible via dual-axis readout of multiple spin resonances for a single hBN defect.

## Results

### An $S=1$ system with dynamic range at high magnetic field

Figure 1 presents the carbon-related hBN spin defects investigated in this work, represented schematically in Fig. 1a. This defect is grown into wafer-scale multilayer (30-nm thick) hBN via metal-organic vapour phase epitaxy (MOVPE) in the presence of triethylboron[37,39–41]. This results in individually addressable, bright spin defects (saturation count rates measured in the range 5–600 kcps, see Supplementary Note 2) that are resolved via scanning confocal microscopy with 532-nm illumination (see Fig. 1b). Figure 1c presents an example photoluminescence (PL) spectrum measured at room temperature, showing zero-phonon line emission at ~2.1 eV accompanied by lower energy phonon side band typical of visible hBN defects[37,41–44]. Figure 1d presents the defect electronic structure, with spin-triplet ground and optically excited states and a spin-singlet metastable state[39]. Relaxation from the optically excited state to the ground-state manifold can occur radiatively or non-radiatively through a sequence of spin-dependent direct and reverse intersystem crossing events that are responsible for optical spin initialisation. The ground-state spin triplet gives rise to three possible paramagnetic transitions between the three spin sublevels, labelled $f_{A-C}$. We study the properties of the ground-state spin via ODMR. Our experimental setup consists of a home-built confocal microscope equipped with a permanent magnet that can be moved in proximity and orientation with respect to the device, enabling a magnetic field up to 140 mT. A coil

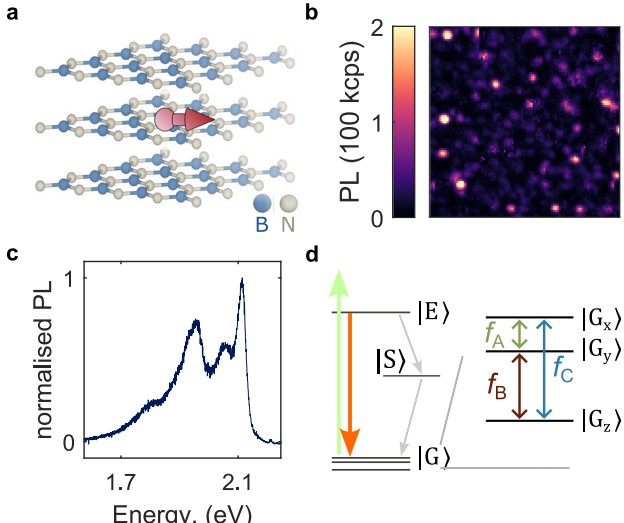

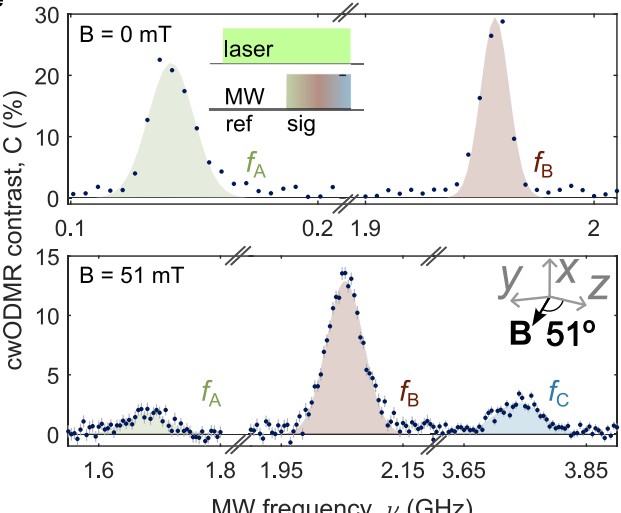

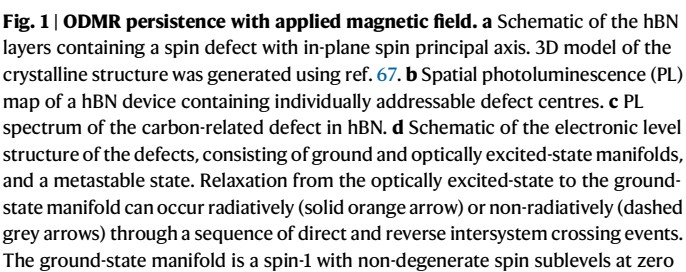

**Fig. 1 | ODMR persistence with applied magnetic field. a** Schematic of the hBN layers containing a spin defect with in-plane spin principal axis. 3D model of the crystalline structure was generated using ref. 67. **b** Spatial photoluminescence (PL) map of a hBN device containing individually addressable defect centres. **c** PL spectrum of the carbon-related defect in hBN. **d** Schematic of the electronic level structure of the defects, consisting of ground and optically excited-state manifolds, and a metastable state. Relaxation from the optically excited-state to the ground-state manifold can occur radiatively (solid orange arrow) or non-radiatively (dashed grey arrows) through a sequence of direct and reverse intersystem crossing events. The ground-state manifold is a spin-1 with non-degenerate spin sublevels at zero

magnetic field. Spin-resonance transitions between each of the three spin sublevels are possible, giving rise to three spin-resonance signatures, labelled $f_{A,B,C}$ in ascending energy. **e** cwODMR spectra measured at 0 mT (top panel) and 51(1) mT (bottom panel), showing three spin transitions between the spin sublevels of an $S=1$ system. Blue circles are measured mean values, with grey error bars indicating the standard error of the mean. Shaded regions are fits to the data using a Gaussian peakshape. The inset in the top panel presents the pulse sequence used for detecting cwODMR, whereas the inset in the bottom panel presents the direction of the magnetic field with respect to the defect's symmetry axes.

in the vicinity of the device delivers microwaves to the hBN defect[37,39].

Figure 1e (top panel) shows the room-temperature ODMR spectrum for an hBN defect at 0 mT, where the microwaves were applied in the range 0.01–3 GHz. The inset shows the measurement sequence for detecting the continuous wave (cw) ODMR contrast, defined as the relative change in PL under 532-nm illumination induced by the presence of microwaves ($C = (\mathrm{PL_{sig}} - \mathrm{PL_{ref}})/\mathrm{PL_{ref}}$). For this defect, we observe two ODMR resonances, at 0.140(2) and 1.957(1) GHz, with comparable saturated cwODMR contrast of 22(5)% and 30(2)%, (see Supplementary Note 1 for zero-field contrast statistics for a range of defects)[39]. We assign the ODMR resonances to the transitions of the $S = 1$ system based on a Hamiltonian of the type,

$$H = H_{ZF} + H_{ZE}, \qquad (1)$$

$$H_{ZF} = DS_z^2 + E(S_x^2 - S_y^2), \qquad (2)$$

$$H_{ZE} = \frac{\gamma_e}{2\pi} \mathbf{B} \cdot \mathbf{S}, \qquad (3)$$

where $H_{ZF}$ is the zero-field splitting term, $H_{ZE}$ is the Zeeman term, $D$ and $E$ are the zero-field splitting parameters that define the defect's $x$, $y$, $z$ principal axes in units of Hz, $\mathbf{S}$ is the $S = 1$ operator with cartesian components $S_{x,y,z}$, $\gamma_e$ is the electron gyromagnetic ratio, and $\mathbf{B}$ is the applied magnetic field. In the absence of applied magnetic field, we only need to consider the $H_{ZF}$ term with eigenenergies 0, $D{-}E$, and $D + E$.

The magnitude of the transverse zero-field splitting $|E|$, relative to $|D|$, is a measure of the rhombicity, or low symmetry, of the spin density of the system[45]. In systems where $|E|$ is low compared to the linewidth (i.e., for the NV centre in diamond and the $V_B^-$ defect in hBN), overlapping resonances are observed at zero field, corresponding to transitions between $|m_s = 0\rangle$ and the near-degenerate $|m_s = \pm 1\rangle$, where $m_s$ denotes the spin projection along the defect's $z$ axis[46]. In such systems, the spin transitions give partial information about the vector of the external magnetic field—while the projection of the field along the $z$ axis (polar dependence) can be determined, the azimuthal direction cannot. In contrast, in the case of low-symmetry $S = 1$ systems, where $|E| \neq 0$, three transitions may arise between the three spin sublevels at zero field indicated in Fig. 1c[45,47–50]. In this case, the transverse zero-field splitting term $E(S_x^2 - S_y^2)$ hybridises $|m_s = \pm 1\rangle$, relaxing the selection rules for transitions between them[48]. The zero-field spin eigenstates are then given by $|G_z\rangle = |m_s = 0\rangle$, $|G\rangle_x = (|m_s = +1\rangle - |m_s = -1\rangle)/\sqrt{2}$, and $|G_y\rangle = (|m_s = +1\rangle + |m_s = -1\rangle)/\sqrt{2}$. We assign the zero-field resonances shown in Fig. 1e (top) to the transition between $|G_x\rangle$ and $|G_y\rangle$ ($f_A$), and $|G_z\rangle$ and $|G_y\rangle$ ($f_B$), where $|D| = 2.027$ GHz and $|E| = 70$ MHz for this defect. Previous work on this defect type has reported the presence of all three transitions, but $f_A$ was outside of the studied measurement range at zero field[39].

Figure 1e (bottom panel) presents the ODMR spectrum for the same defect under 51-mT magnetic field applied in the plane of the hBN layers. At this field, all three spin transitions are visible in the spectrum, with $C(f_A) = 1.8(2)\%$, $C(f_B) = 12.9(5)\%$, and $C(f_C) = 2.7(3)\%$, where $C(f_i)$ is the contrast of the i-th transition. We determine that the field vector is at 51(1)° from the defect $z$ axis, parallel to the $yz$ plane, from field-dependent measurements. This means that the defect's $y$ and $z$ axes are parallel to the plane of the hBN layers, and the $x$ axis is out of the plane. Despite the high off-axis applied field, we observe that the ODMR resonance is not quenched. This is in contrast to what is seen for the NV centre, where a magnetic field -10 mT misaligned to the defect's quantisation axis quenches the ODMR resonances due to degradation of the spin initialisation mechanism[17,51].

## Photodynamics of the carbon-related hBN spin

For optically active spin defects, the ODMR contrast is dependent on the degree of spin initialisation arising from the optical cycle. To understand the remarkably high ODMR contrast and its retention with off-axis field for the hBN defects, we investigate the optical rates of the system by setting up a series of rate equations describing the transfer of population between the electronic states for the model shown in Fig. 2a, in the absence of a magnetic field. Across the defects we study, we observe the magnitude of the saturated zero-field ODMR contrast across the three spin resonances typically follows: $C(f_A) = C(f_B) > C(f_C)$ with defect-to-defect variation in overall magnitude[39] (see Supplementary Fig. 2). This observation is in line with an optical defect that shows variable intersystem crossing (ISC) rates, consistent with the variation we see in bunching timescales in second-order autocorrelation ($g^{(2)}(t)$) experiments[37]. The non-equal ODMR contrast of $f_B$ and $f_C$ indicates strong spin selectivity of the ISC at zero-field, as is observed for other low symmetry $S = 1$ systems[47,48,52]. In our kinetic model, we hold $k_{E_x \to S_0} = k_{E_z \to S_0} \neq k_{E_y \to S_0}$ and $k_{S_0 \to G_x} = k_{S_0 \to G_z} \neq k_{S_0 \to G_y}$ in order to restrict the number of fitting parameters, but note that some defects are best described by $k_{E_x \to S_0} \neq k_{E_z \to S_0} \neq k_{E_y \to S_0}$ ($k_{S_0 \to G_x} \neq k_{S_0 \to G_z} \neq k_{S_0 \to G_y}$).

We estimate the optical rates for a second single defect at zero field via a global fit to the combined results of the second-order autocorrelation ($g^{(2)}(t)$, Fig. 2b), spin initialisation and relaxation measurements (Fig. 2c, d), with the cwODMR magnitude and optical saturation parameters acting as experimental bounds. The pulsed ODMR sequences are illustrated in the insets of the respective figures, where the microwave pulses are $\pi$ pulses calibrated via Rabi experiments on resonance with $f_B$ (see Supplementary Note 3). Figure 2b shows the background-corrected $g^{(2)}(t)$ measurement for this defect (see Supplementary Note 4 for details on the background correction procedure). The horizontal (time) axis is presented in linear scale between −30 and 30 ns, where we can see the characteristic anti-bunching dip at $t = 0$. For $|t| > 30$ ns, we present the time axis in log scale. The hBN defects show significant bunching behaviour, indicative of the presence of a long-lived metastable state, which only subsides after -100 μs. Similar trends have been reported for various types of hBN emitters[37,53–57].

In Fig. 2b–d the red curves are the result of a global fit of a $S = 1$ optical model to the experimental data, and the grey shaded region reflects the confidence regions for the fits. Table 1 presents the corresponding rates extracted from this fit (see Supplementary Note 5, for details on model, fitting procedure, and uncertainties). We note in our analysis we also considered a model with spin-singlet ground and optically excited states and spin-triplet metastable state, but this model fails to capture the observed behaviour (see Supplementary Note 6). For this defect, the global fit reveals comparable magnitudes for the radiative ($\Gamma_{E \to G} = 163$ MHz) and non-radiative ($k_{E \to S_0} = \sum_{i = x, y, z} k_{E_i \to S_0} = 200.8$ MHz) decay rates from the optically excited state, and strongly spin-selective direct and reverse intersystem crossing ($k_{E_y \to S_0}/k_{E \to S_0} = 0.946$ and $k_{S_0 \to G_y}/k_{S_0 \to G} = 0.994$).

We repeat this procedure for five defects with the same zero-field splitting resonance and find that, while the magnitude of the radiative and intersystem crossing rates are broadly similar across defects, there is significant variation in the ratio of spin-dependent intersystem crossing rates ($k_{E_y \to S_0}/k_{E \to S_0} = 0.49{-}0.95$, $k_{S_0 \to G_y}/k_{S_0 \to G} = 0.82{-}0.99$). This provides an explanation for the defect-to-defect variation (from <1% to 95%) in the magnitude of the saturated ODMR contrast[39] (see Supplementary Note 7 for extended data from which individual rates are extracted). Figure 2e shows the interdependence of the cwODMR contrast on the spin-selectivity of the direct ($k_{E \to S_0}$, vertical axis) and reverse ($k_{S_0 \to G}$, horizontal axis) intersystem crossing rates. The 2D map presents the simulated cwODMR contrast of $f_B$, where the rates indicated in the axes are varied while all remaining rates are kept constant at the values presented in Tab.1. The colour represents the amplitude of cwODMR contrast predicted by the model, with red

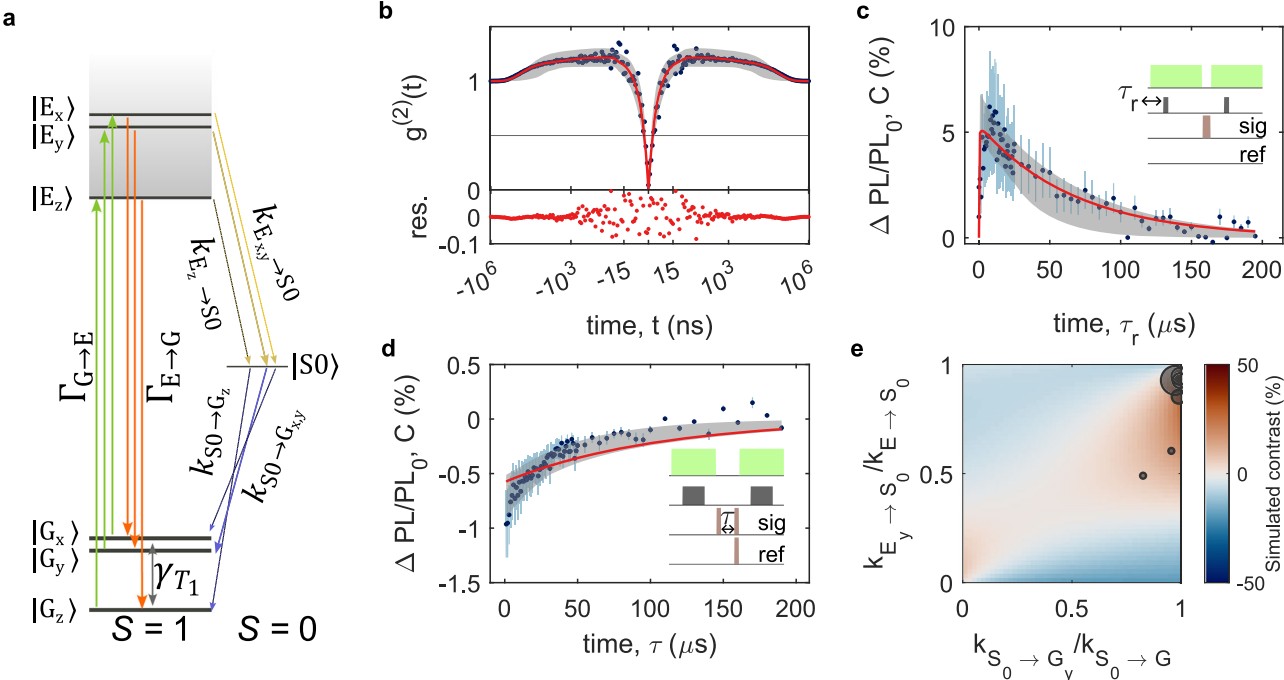

**Fig. 2 | Optical and spin dynamics of carbon-related hBN defects. a** Model used to fit the results of experiments in (**b**–**d**), including a spin-1 ground and optically excited states and a singlet metastable state. Optical excitation and radiative recombination processes are spin-conserving at zero magnetic field. **b**–**d** Blue circles correspond to the mean of measured data, light blue error bars indicate one standard deviation of the measured data, red curves correspond to a global fit of the model to the experimental data, and the grey shaded region corresponds to model predictions with <3× the minimum fit error. The insets present the pulse protocols used in the measurements, with optical (green blocks) and microwave drive (red blocks) pulses and readout time (grey block). **b** (top) Background-corrected second-order autocorrelation ($g^{(2)}(t)$). (bottom) Residuals of the fit of the model to the data. **c** Spin-dependent optical initialisation. Blue circles are the mean value of the contrast measured for various delay times $\tau_r$. **d** Modified spin-relaxation experiment. The signal experiment probes the PL when we apply two microwave $\pi$ pulses, before and after a delay time $\tau$ between the two optical pulses. The reference experiment probes the PL when a single microwave $\pi$ pulse is applied at the end of $\tau$. **e** Simulated ODMR contrast as a function of the spin-selectivity of the direct ($k_{E\to S_0}$) and reverse ($k_{S_0\to G}$) intersystem crossing rates. The size of the black circles represents cwODMR contrast of different defects, positioned according to their relative rates extracted from fits to second-order autocorrelation and pulsed ODMR experiments. The largest circle corresponds to a 30% contrast.

**Table 1 | Model parameters**

| Rate | $\Gamma_{G\to E}$ | $\Gamma_{E\to G}$ | $k_{E_x\to S_0}$ | $k_{E_y\to S_0}$ | $k_{E_z\to S_0}$ | $k_{S_0\to G_x}$ | $k_{S_0\to G_y}$ | $k_{S_0\to G_z}$ | $\gamma_{T_1}$ | Contrast |
|---|---|---|---|---|---|---|---|---|---|---|
| Unit | kHz/µW | MHz | MHz | MHz | MHz | kHz | kHz | kHz | kHz | % |
| | $0.92^{0.98}_{0.17}$ | $163^{243}_{79}$ | $5.4^{125}_{1}$ | $190^{376}_{127}$ | $5.4^{125}_{1}$ | $2^{1E5}_{0}$ | $675^{1E6}_{146}$ | $2^{1E5}_{0}$ | $3.2^{7.6}_{2.8}$ | 12 |

Summary of key parameters obtained from fitting the model of Fig. 2a to the data in Fig. 2b–d.

(blue) regions indicating positive (negative) contrast. The black circles show the experimental cwODMR contrast for each defect we measured (where the size represents the magnitude of cwODMR contrast, see Supplementary Note 8 for the raw spectra), positioned on the map as a function of the determined rates for each defect. The rates extracted using the procedure outlined above for the hBN defects cluster in the top right of the 2D plot, showing that these defects are characterised by strong spin-selectivity in both direct and reverse intersystem crossing processes. The strong spin-dependence of ISC means that spin mixing requires a larger applied magnetic field in order to disrupt the optical spin initialisation mechanism[17], giving rise to a large magnetic-field dynamic range for the hBN sensor.

To explore the optical response of the hBN defects to arbitrary oriented applied magnetic field, we perform angular-dependent ODMR (Fig. 3). Figure 3a, c show the dependence of cwODMR central frequencies (top panel) and normalised cwODMR contrast of $f_A$–$f_C$ (bottom panels), on the orientation of 51-mT magnetic field in the $yz$ (a) and $xy$ (c) planes, for the same defect that is presented in Fig. 1, where the contrast is normalised by the zero-field cwODMR contrast of $f_B$ (experimental data is shown by circles). Interestingly,

we see that applied field along the defect $y$ axis (indicated by 90 degrees in Fig. 3a, c) preserves the zero-field contrast. Meanwhile, the cwODMR contrast of $f_A$ ($f_B$) is completely (partially) suppressed as the magnetic field is rotated towards the $z$ axis, while the cwODMR contrast of $f_C$ increases (Fig. 3a). We note that the sharp dip in contrast of $f_A$ and $f_B$ when the field is applied directly along the $y$ axis is reproducible, but we have not identified its origin. Rotation of the applied field in the $xy$ plane away from the $y$ axis leads to a slower suppression of the cwODMR contrast of both $f_A$ and $f_B$, with a correspondingly slower increase of the cwODMR contrast of $f_C$ (Fig. 3c).

The experimental data in this figure (circles) is shown alongside the results of the modelled cwODMR contrast (solid curves (Fig 3a, c) and 2D colour maps (in Fig 3b, d)) for this defect, determined from the zero-field rates obtained in the analysis above, where we include the effect of magnetic field by introducing the Zeeman term to the spin Hamiltonian ($H_{ZE}$). We determine the magnetic-field-dependent intersystem crossing rates from a statistical average of the zero-field rates, such that $k_{ij}(\mathbf{B}) = \sum_{p,q}|a_{ip}|^2|a_{jq}|^2 k^0_{pq}$, similar to the approach taken by Epstein et al. and Tetienne et al. for the NV centre in diamond[17,58]. Here,

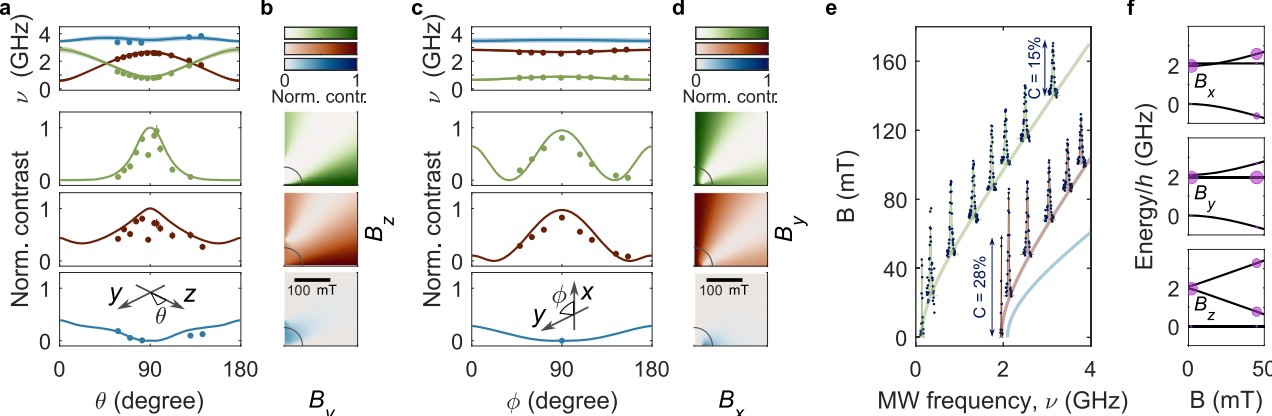

**Fig. 3 | Magnetic-field orientation and amplitude dependence of cwODMR.**
**a** Angular magnetic-field dependence of cwODMR frequency (top panels) and contrast of resonances $f_A$ to $f_C$ (top to bottom), normalised by the zero-field cwODMR contrast of the $f_B$ resonance, with 50-mT bias magnetic field applied in the $yz$ plane. Data are presented as circles, with colour coding according to the inset of Fig. 1d, and curves indicate the cwODMR contrast simulated using the model of Fig. 2 and fit parameters presented in Tab. 1. The inset indicates the direction of rotation of the bias magnetic field. **b** Calculated contrast of each resonance as a function of $B_y$ and $B_z$. **c** as (**a**), but with 50-mT bias field applied in the $xy$ plane of the defect ($\phi$ varies with fixed $\theta = 85°$). **d** Calculated contrast of each resonance as a function of $B_x$ and $B_y$. **e** Persistence of saturated cwODMR contrast for field applied along the defect $y$ direction ($\phi = 90°$, $\theta = 85°$), shown up to 140 mT. The solid curves represent the transition frequencies of $f_A$ (green), $f_B$ (red), and $f_C$ (blue) resonances as a function of $B_y$ amplitude. The measured cwODMR contrast as a function of MW frequency is represented by blue circles. **f** Evolution of spin eigenstates of the Hamiltonian Eq. (1) with applied magnetic field along the $x, y, z$ axes of the defect, from top to bottom. Calculated amplitudes of the optically initialised population of each spin sublevel are indicated by the size of the purple circles.

$k_{pq}^0$ are the zero-field direct and reverse spin-dependent intersystem crossing rates; the coefficients $a_{ip}$ can be obtained by comparing the zero-field eigenstates ($|p(0)\rangle$) to the eigenstates of the Hamiltonian at a field ($|i(\mathbf{B})\rangle$), such that $|i(\mathbf{B})\rangle = \sum_p a_{ip}|p(0)\rangle$. In the absence of spectroscopic information about the excited-state zero-field splitting configuration, we assume equal zero-field splitting parameters in ground and optically excited states, and this assumption has no significant implication on the findings of this work (see Supplementary Note 9).

Figure 3b, d shows the calculated normalised contrast of each cwODMR resonance for this defect as a function of $B_y$ and $B_z$ ($B_x$) up to 200 mT with $B_x$ ($B_z$) fixed at 0 mT. These simulations show that the behaviour outlined above persists for a wide range of bias magnetic field values, with limited regions showing complete quenching of all three cwODMR resonances. Figure 3e presents the experimental cwODMR spectra as a function of $B_y$ amplitude up to 140 mT, showing that for this class of defects, contrast is preserved for an applied field along the defect's $y$ axis. Importantly, this data reveals a spin system with multiple quantisation axes, where optical initialisation of each spin transition has a different dependency on the orientation of the applied field[45].

Figure 3f presents the evolution of the ground-state spin eigenstates for the hBN defect system under applied magnetic field in the $x, y, z$ direction (top to bottom panels). The purple circles represent the simulated optically initialised population, calculated based on the model above and using the representative rates of Tab. 1. As observed experimentally, in the zero-field limit, the kinetic model predicts the system is initialised into the $|G_y\rangle$ state, giving rise to strong $f_A$ and $f_B$ and weak $f_C$ (Fig. 1). Magnetic field applied along the defect $y$ axis (middle panel) mixes $|G_x\rangle$ and $|G_z\rangle$, preserving the zero-field character of $|G_y\rangle$, thus retaining the zero-field spin initialisation and ODMR contrast. Conversely, applied field along $x$ ($z$) mixes $|G_y\rangle$ and $|G_z\rangle$ ($|G_x\rangle$), redistributing the zero-field initialised population and modifying the saturated cwODMR contrast of each resonance with respect to their zero-field values. In total, the experimental data and kinetic model both reveal a multi-axis sensor with optical initialisation dynamics that enable operation under strong off-axis field.

## Discussion

DC sensitivity is given by the relationship $\eta_{DC} = \alpha \frac{1}{\partial \nu_i/\partial B} \frac{\Delta \nu}{C\sqrt{PL_0}}$, where $\alpha$ is a prefactor associated with the cwODMR lineshape ($\alpha = \sqrt{e/(8 \log 2)}$ for a Gaussian lineshape), $\partial \nu_i/\partial B$ is the resonance frequency dependence on magnetic-field amplitude, $\Delta \nu$ is the cwODMR resonance full width at half maximum, $C$ is the contrast and $PL_0$ is the brightness of the defect in the absence of microwaves, where collection losses are accounted for[11,59]. For the defect shown in Figs. 1 and 3, the DC sensitivity is 15 µT Hz$^{-1/2}$, and for the defect in Fig. 2 it is 10 µT Hz$^{-1/2}$, with experimental sensitivity ranging from 1.5 µT Hz$^{-1/2}$ to 1.5 mT Hz$^{-1/2}$ for defects we have studied (see Supplementary Note 10). This wide range is related to the range in ODMR contrast and brightness, with the best defects in hBN performing similarly to typically determined sensitivity for shallow NV centres in terms of absolute sensitivity[12].

### Vectorial magnetic-field sensing enabled by the low-symmetry spin

Full vectorial sensing enables unambiguous determination of three linearly independent target magnetic field components. In Fig. 4 we present an experimental protocol that can determine three components of a target-field vector with the hBN defect, consisting of measuring ODMR under two bias fields. Figure 4a presents a schematic of the proposed measurement protocol. Figure 4b shows the unit vectors associated with the 0.1-mT target magnetic field **T** (grey) with components $T_{x,y,z}$ and the two arbitrarily oriented bias fields ($\mathbf{B_1}$ in light blue and $\mathbf{B_2}$ in pink), all non-collinear with each other or with any of the defect's axes. Figure 4c, d shows the calculated cwODMR spectra with $\mathbf{B_1}$ and $\mathbf{B_2}$, respectively, and the shifts induced by the presence of the target magnetic field. Each component of the target-field ($T_x$, $T_y$, $T_z$, red, blue, and green dashed curves, respectively) induces a significant and distinct shift to each ODMR resonance, enabling determination of all three target-field components unambiguously. This is shown in more detail in Fig. 4e. Here, the grey vector represents the 0.1-mT target magnetic field, and the shaded pink and light blue clouds indicate the possible solutions for the target-field vector from a 1-second cwODMR measurement with $\mathbf{B_1}$ and $\mathbf{B_2}$, respectively. With $\mathbf{B_1}$ in the $yz$ plane, $T_y$ and $T_z$ can be determined, but the measurement gives no

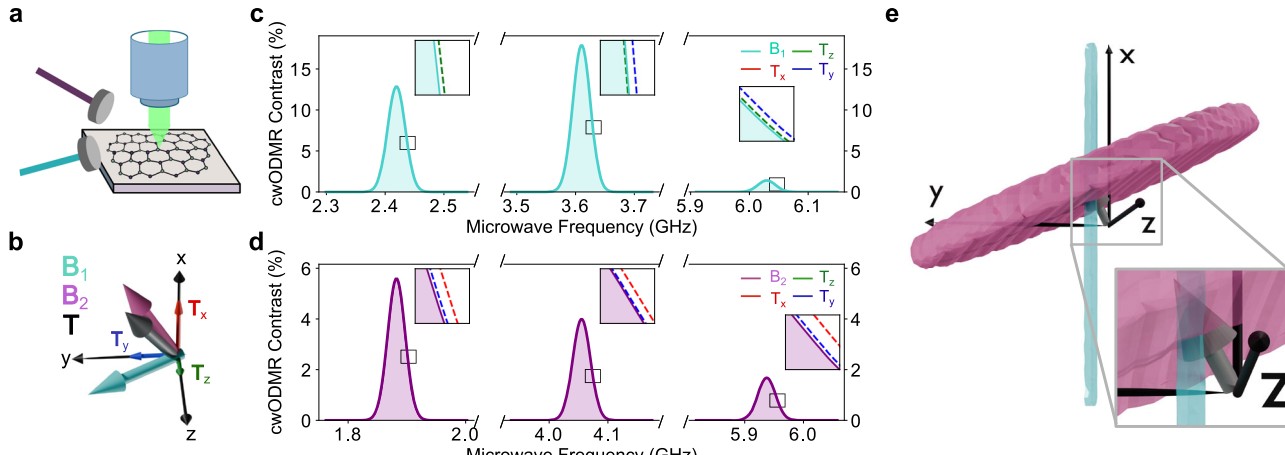

**Fig. 4 | Proposed experiment for vectorial magnetometry. a** Experimental setup, consisting of a typical confocal microscope with the possibility of applying a bias magnetic field in two different orientations. **b** Unit vectors indicating the orientations of $B_1$ and $B_2$, two 100 mT bias magnetic fields we use in this simulation, as well as the orientation of a 0.1-mT target magnetic field T with components $T_{x,y,z}$. **c, d** cwODMR spectra with $B_1$ and $B_2$, respectively. We monitor the change in contrast of each resonance due to the presence of the target magnetic field (shown explicitly in the inset) to gain information about each of its three vectorial components. **e** Ambiguity in determining the target magnetic field given a 1-second measurement of the contrast at the slope of each resonance. The grey arrow represents the target magnetic field. The clouds represent field configurations that cannot be distinguished from the target field under $B_1$ (light blue cylinder) and $B_2$ (pink dome).

information on $T_x$ component. This is reflected by the cylindrical light blue cloud with its long axis parallel to the $x$ axis. Alternatively, a measurement with $B_2$ in the $xy$ plane provides information on the target-field orientation within a dome normal to the bias field, thus gaining information on $T_x$. Combined, the two measurements performed with $B_1$ and $B_2$ can determine the three components of the target-field vector, represented by the overlap in these two distributions (see inset). In conclusion, the hBN defect offers a route to full vectorial-field mapping of a target field.

## Outlook

Our results present a new candidate for nanoscale quantum sensing with intrinsic characteristics that make it attractive for nanoscale magnetometry. We show that the photodynamics of this system gives rise to an atomic sensor that can operate under a transverse magnetic field of over 150 mT, with DC sensitivity of ~1μT/Hz$^{-1/2}$, a degree of sensitivity that is comparable to what is typically achieved for shallow NVs[12,60]. Further, the defects can be grown into multilayers only a few-nm thick, and the 2D host enables simple integration into 2D heterostructures, as well as established tip-based sensing approaches, which opens routes to achieving well below 20-nm spatial resolution with this platform. In addition, the hBN host is inert in biological media[61], inexpensive to produce, and can be grown to scale.

We postulate that the 2D host material may offer future advantages for optimal tuning of the dynamics of defects, in situ. Interestingly, our kinetic modelling and ODMR results across defects of this type show that the magnitude and ratio of optical rates depend on the defect in question, despite a well-defined ground-state electronic structure (see Supplementary Note 7 for statistics on radiative and non-radiative decay rates across defects). We predict that the excited-state dynamics in this system are highly sensitive to local strain and electric field, due to being embedded in a 2D material that can be highly strained. Despite being a source of inhomogeneous behaviour, this feature could be harnessed as a pathway to control and enhance contrast of individual emitters via strain or electric field tuning, for example. However, hBN defects present an immature technology where multiple emerging engineering challenges must be addressed before it becomes a scalable technology. These include developing approaches to control the defect-to-defect inhomogeneity with regard to brightness and ODMR contrast. While every hBN defect that displays the $D$ ~ 1.9 GHz spin resonance shows similar spin coherence properties, the overall contrast and brightness depend on the defect, as we discuss in this report. In addition, the relatively low spin coherence times compared to isotopically purified diamond will place an upper limit on the sensitivity for sensing.

Finally, these results pertain mainly to the performance of this system for DC sensing. However, the spin properties of this hBN defect, including microsecond spin coherence at room temperature that can be accessed via dynamical decoupling protocols[39], open routes for exploring the system for AC sensing. hBN defects are stable in aqueous environments[62], which may lend the platform to magnetometry, or ensemble based sensing in biological systems in the future[63–65].

## Methods
### Materials
Multilayer hBN was grown by metal-organic vapour phase epitaxy on sapphire substrates and subsequently transferred to Si/SiO$_2$ using water-assisted self-delamination[40]. Triethylboron and ammonia were used as boron and nitrogen precursors during MOVPE, where the flow rate of triethylboron has been shown to be correlated to the incorporation of carbon and the observation of visible emitters in the resulting hBN[41].

### Confocal PL microscopy
PL measurements are conducted at room temperature on two home-built free-space confocal microscopy setups. In both cases, experimental hardware is connected to a data acquisition card (National Instruments, PCIe6323), controlled via an open-source Python suite Qudi[66]. We use a 532-nm continuous wave laser (Ventus 532, Laser Quantum), split into several excitation lines using a beamsplitter, where each path is directed to a different setup. In each line, the laser passes through an acousto-optic modulator (AA Optoelectronics) for intensity control and pulse generation. The laser is filtered using a 532-nm laser line filter (Thorlabs, FL532-3) after out-coupling into the free-space microscopy setup. The beam passes through a beamsplitter (Thorlabs, 90:10 R:T), with the reflected beam providing sample excitation, and the intensity of the transmitted beam is monitored using a photodiode, completing a feedback loop that allows laser-power control in conjunction with the AOM. Confocal scanning of the

sample is enabled by a scanning mirror (Physik Instrumente, S-334.2SL), and a ×100 0.9 NA air objective (Nikon Instruments). The emitted light is collected by the same objective and optical path, and coupled into a single-mode fibre (Thorlabs, SM600) after passing through two 550-nm long-pass filters (FEL550, Thorlabs) to remove the laser light. The collected light is sent to either a single-photon avalanche photodiode (SPCM-AQRH-14-FC, Excelitas Technologies) or to a charge-coupled-device-coupled spectrometer (Acton Spectrograph, Princeton Instruments).

### Optically detected magnetic resonance

ODMR measurements are conducted using confocal microscopy as described above. Microwaves are produced using an RF signal generator (Stanford Research Systems DC to 4.05 GHz Signal Generator or Marconi Instruments 10 kHz to 2.4 GHz Signal Generator), amplified (ZHL-42 W+, 0–4.2 GHz, 30–35 dB, MiniCircuits), and delivered to the sample by a homemade copper loop antenna placed between the objective lens and the hBN multilayer, with the confocal spot approximately aligned to the centre of the loop to allow optical access.

During cwODMR, the 532-nm excitation is continuously applied, whilst the intensity of the microwaves is modulated using a 140-Hz square wave, and the microwave frequency is modulated at 7 Hz. The PL count rate is monitored as a function of microwave amplitude and frequency to calculate ODMR contrast, representing the fractional change in PL counts upon the application of microwave drive ($C = (PL_{sig} - PL_{ref})/PL_{ref}$). The contrast is calculated per datapoint, and finally averaged for each microwave frequency position across all measurement sweeps.

For pulsed ODMR measurements, we use a pulse streamer (Swabian 8/2) to control a series of switches (MiniCircuits ZYSWA-2-50-DR+). The switches allow the generation of square-wave laser and microwave pulses by modulating the AOM trigger level and the connection between the RF source output and the antenna, respectively. A switch also controls the signal readout duration. The pulse sequences used during measurements are described in detail in the text, the Supplementary Note 5.

Angle-resolved magnetic field is applied using a dual-axis home-built mount, consisting of two rotation stages (Thorlabs) and a permanent magnet on a custom mount. Zero-field measurements are conducted without shielding, in the Earth's magnetic field.

### Hanbury-Brown Twiss interferometry

Intensity autocorrelation measurements are conducted using Hanbury-Brown-Twiss interferometry. The fluorescence collection fibre is connected to a 50:50 fibre beamsplitter, with each end coupled into a single-photon avalanche photodiode. The PL counts at each photodiode are monitored by a time-to-digital converter (quTAU, qutools) with 81-ps resolution.

## Data availability

All data needed to evaluate the conclusions in the paper are present in the article or its Supplementary Notes. All datasets are available from the corresponding authors upon request. Source data files are provided with this paper. Source data are provided with this paper.

## Code availability

All codes used to generate the figures and analyse the data are available from the corresponding authors upon request.

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

## Acknowledgements

We thank Toby Mitchell for technical assistance, and Viktor Ivády, Sam Bayliss, and Qiushi Gu for helpful discussions. We acknowledge support from the ERC Advanced Grant PEDESTAL (884745) and the Office of Naval Research Global (N62909-22-1-2028). C.M.G. acknowledges support from the Netherlands Organisation for Scientific Research (NWO 019.221EN.004, Rubicon 2022-1 Science) and from the Herchel Smith Postdoctoral Fellowship Fund. S.E.B., S.A.F., and O.F.J.P. acknowledge funding from the EPSRC Centre for Doctoral Training in Nanoscience and Nanotechnology (NanoDTC, grant no EP/S022953/1). C.L.C. acknowledges funding from the EPSRC Doctoral Training Programme and the Vice-Chancellor Award. D.L. acknowledges scholarships from the Skye Foundation and the Cambridge Trust. C.L., I.A., and H.H.T. acknowledge funding from the Australian Research Council, through grants CE200100010 and FT220100053. H.L.S. acknowledges a Royal Society fellowship.

## Author contributions

C.M.G. and H.L.S. conceived the study. S.E.B. conducted the experiments, with the help of C.L.C., S.A.F., O.F.J.P., D.K.L., and X.D. C.M.G., S.E.B., and H.L.S. analysed the data. C.M.G. developed the microscopic model. C.M.G., S.E.B., C.L.C., S.A.F., O.F.J.P., A.J.R., M.A., and H.L.S.

participated in regular discussions of the results. S.A., C.L., I.A., and H.H.T. provided the material and devices used. H.L.S. and M.A. supervised the work. All authors contributed to the drafting of the manuscript, with leadership from C.M.G., S.E.B., and H.L.S.

## Competing interests

C.M.G., S.E.B., H.L.S., and M.A. are listed as inventors in UK patent application number 2409550.7, with application status listed as pre-PCT filing. The patent subject is an operation of a nanoscale sensor for vectorial magnetometry based on a single defect in the hBN material platform. All other authors declare no competing interests.
