## [Transparent Peer Review file · Nature Communications]

A single spin in hexagonal boron nitride for vectorial quantum magnetometry

Corresponding Author: Dr Carmem Gilardoni

Version 0:

Reviewer comments:

Reviewer #1

(Remarks to the Author)

A single spin in hexagonal boron nitride for vectorial quantum magnetometry

Carmem M. Gilardoni ‡,1, * Simone Eizagirre Barker ‡,1 Catherine L. Curtin,1 Stephanie A. Fraser,1 Oliver. F.J. Powell,1,2 Dillon K. Lewis,1 Xiaoxi Deng,1 Andrew J. Ramsay,2 Chi Li,3,4 Igor Aharonovich,3,4 Hark Hoe Tan,5 Mete Atature,1 and Hannah L. Stern6,†

Herewith, I am submitting my reviewer comments for the above-mentioned manuscript which is under consideration to be published in Nature Communications.

The article is about magnetometry with single spins in hexagonal boron nitride. This is an interesting topic and an interesting alternative to magnetometry with NV centers. What is particularly interesting in this system is that it is a multi-axis spin system for vectorial nanoscale magnetometry. Another advantage seems to be that the sensor also works at higher magnetic fields.

Overall, the article is well written and clear and can be published after minor revisions.

“Our results present a new candidate for nanoscale quantum sensing that has the potential to extend the technique to new systems across condensed matter physics, biology and biomedical science.” Is there any evidence for this? In order to be useful for biological and medical applications, there is a relatively high brightness needed. (we typically work in biological systems with hundreds of NV centers in conventional patient samples, anything that is less bright than several tens of NVs wouldn't be visible in a medical sample) Do I remember correctly that these defects are much less bright than NV centers? If that's true, I would say the claim that it is for biological and biomedical applications is overly optimistic. (that doesn't mean the material isn't promising for other applications) EDIT: below it says “typical saturation count rates in the range 5-200 kcps” so this is indeed very low when compared to NV centers.

“This is in stark contrast to what is seen for the NV centre, where a magnetic field ~ 10 mT misaligned to the defect's quantization axis quenches the ODMR resonances due to degradation of the spin initialisation mechanism” this is true. But as it stands now the article is a bit overly critical with NV centers. The overall sensitivity of NVs is still much higher and brightness is a real issue in boron nitride. There should be a more balanced discussion which includes the downsides of the boron nitride material.

Reviewer #2

(Remarks to the Author)

Gilardoni et al performed cw ODMR studies of carbon-related spin defects in hBN. They discover that ODMR resonances are still detectable even when the field is applied off-axis. They assign this to low symmetry of the defect's spin density. They further simulate achievable magnetic-field sensitivity using these defects.

Spin defects in hBN have attracted much attention recently due to their high ODMR contrast at room temperature. Applying

them in quantum sensing seems like a logical next step. The manuscript is well written and the results are clearly presented. However, I have reservations towards its publication in Nature Communications. ODMR features of this type of defects have been reported by several groups, including some of the co-authors on this manuscript. The quantum sensing discussion, though provides a new angle, were only performed using simulations rather than experimental demonstrations. Overall, I find the novelty/potential impact of this work to be limited.

I also list a few comments that the authors may consider addressing to further improve the quality/completeness of the manuscript:

- The authors attribute the weak field orientation dependence of the ODMR signals to low symmetry of spin densities. I find this argument arbitrary and lacking proof. The authors should consider connecting this observation to the atomic structure of the defects, and if possible, perform DFT calculations to support this major claim.
- The global model provides a reasonable fit in fig. 2b, but the fits in fig. 2c and 2d are quite poor, suggesting additional energy levels or dynamics not included in the model. The authors should provide possible explanations for this.

Reviewer #3

(Remarks to the Author)

The manuscript by Gilardoni et al investigates an ODMR-active spin defect in hBN previously discovered by the same group, focusing here on its photodynamics and ODMR contrast as a function of magnetic field orientation, showing experimentally that ODMR contrast is preserved for a wide range of orientations, and predicting that a single defect can allow vector magnetometry. While the results are interesting, I find the claims insufficiently supported by the analysis, and so I cannot recommend publication in Nat Comms at this stage.

My main concern is about the analysis of the photodynamics, which is incomplete and unconvincing in my opinion. The authors chose to apply a global fit to the three normalised curves presented in Fig. 2 in an attempt to validate the model and estimate the parameters, but the fit in Fig. 2d is clearly not good which puts the validity of the model and/or parameters into question. At the very least there should be an uncertainty analysis on the extracted parameters listed in table I, to reflect the poor fitting in Fig. 2d and show which parameters are the most uncertain as a result. Additionally and importantly, I find it strange that the authors do not show and analyse the sig and ref traces that led to Fig. 2c and 2d. Just showing that the difference is well fit by the model (for Fig. 2c) is not enough, the authors also need to show the model explains the individual traces before normalisation. Those can be background subtracted if needed (by running the same sequence away from the defect under study). For Fig. 2c, the raw traces correspond to the PL during the laser pulse, which are dictated in part by the $E \rightarrow S_0$ rates. For Fig. 2d, the raw traces are in effect recovery curves, which should reveal the rates $S_0 \rightarrow G$. Given the authors estimate the rates $S_0 \rightarrow G_{x,z}$ and γ_{T1} to be a few kHz, there should be a slow recovery component on the corresponding time scale (~500 us) in the sig and ref traces underpinning Fig. 2d. Without a proper analysis of the full data set and a discussion of the uncertainties in the model and rates, I don't think the conclusions of the paper are sufficiently justified, given the sensitivity predictions in Fig. 4 are based on the parameters inferred from Fig. 2.

Other comments:

- For Fig. 1e, you write, "We determine the field vector is at 51(1)deg from the defect z-axis, parallel to the yz plane, from field-dependent measurements." -> Please define the y axis. Does the yz plane coincide with the plane of the hBN layers? That should be explained. Also, in the inset of Fig. 1e, the schematic says 52deg rather than 51deg.
- In the main text you write "see Fig. S3 for zero-field contrast statistics for a range of defects" -> I think you meant Fig. S1
- In Fig. 3b,c the ODMR contrast is calculated as a function of magnetic field direction for a constant magnitude of 51 mT. Since a key point of the paper is the extended magnetic field range, I think there should be predictions as a function of the field magnitude, e.g. the ODMR contrast of each resonance could be plotted in a heat map as a function of (B_x, B_y) and another one as a function of (B_y, B_z) to extend Fig. 3b,c.
- Likewise, the sensitivity predictions in Fig. 4 assumes a constant magnitude of 50 mT. I think the angular plots which are hard to read should be complemented by 2D heat maps as a function of (B_x, B_y) and (B_y, B_z) . This would also allow a more meaningful comparison with the NV case beyond just the special case of the ESLAC region. It would also justify the claim in the abstract that "provide sub- $\mu T/rt(Hz)$ magnetic-field sensitivity for both on and off-axis bias magnetic field exceeding 50 mT" (the word "exceeding" lacks justification).
- For the sensitivity calculations, if I understand you took PL_0 based on the rates inferred from Fig. 2 corrected by a factor 0.1 to account for collection losses. At optical saturation, this would mean a photon collection rate of $\Gamma_{E \rightarrow G} = 163$ MHz divided by roughly 2 to account for the ISC and multiplied by the 0.1 collection efficiency, which gives 8.5 Mcounts/s, far above the saturation count rates reported in the SI. There could be many reasons for this discrepancy (e.g. the $\Gamma_{E \rightarrow G}$ decay may have a non-radiative component), but in any case the authors should follow the common practice of using the measured photon rate (i.e. 200 kcounts/s) rather than an unverified prediction. This is especially important since you are making claims about sub- $\mu T/rt(Hz)$ sensitivity in the abstract, which misleadingly suggests the defect outperforms NV.
- The explanation of how measurements with two different bias fields allow full vector magnetometry is not very clear to me. Fig. 4c plots sensitivity but that does not imply there is a unique solution for the vector magnetic field. The uniqueness of the solution needs to be discussed and a simulated set of ODMR spectra for a test vector field (plus two different bias fields) and its reconstruction would help greatly to illustrate the point.

Version 1:

Reviewer comments:

Reviewer #1

(Remarks to the Author)

The authors have addressed the raised concerns sufficiently and the article can be published.

To clarify what I meant in comment 1: I didnt mean to do ensembles of NV centers to increase sensitivity. I do not doubt their claim on the improved sensitivity. My main concern is about brightness. If we use clinical samples and most "normal" biological material, we need several tens of NV center in order to be able to see/find any particles.

Reviewer #2

(Remarks to the Author)

The authors have addressed my comments adequately. I recommend its publication as it is.

Reviewer #3

(Remarks to the Author)

The authors did a good job at addressing my concerns. The fitting procedure and robustness including uncertainty analysis are now sufficiently convincing in supporting the claims. The heat maps in Fig. 3 are helpful. The new Fig. 4 is much clearer in explaining the vector sensing idea. Overall I think the manuscript has greatly improved and I'd be happy to see it published in Nat Comms.

Reviewer #1 (Remarks to the Author):

1. Herewith, I am submitting my reviewer comments for the above-mentioned manuscript which is under consideration to be published in Nature Communications.

The article is about magnetometry with single spins in hexagonal boron nitride. This is an interesting topic and an interesting alternative to magnetometry with NV centers. What is particularly interesting in this system is that it is a multi-axis spin system for vectorial nanoscale magnetometry. Another advantage seems to be that the sensor also works at higher magnetic fields. Overall, the article is well written and clear and can be published after minor revisions.

Reply:

We would like to thank the reviewer for recognising the impact of our work and quality of the submitted manuscript, and their recommendation for publication in Nature Communications.

2. *“Our results present a new candidate for nanoscale quantum sensing that has the potential to extend the technique to new systems across condensed matter physics, biology and biomedical science.” Is there any evidence for this?*

In order to be useful for biological and medical applications, there is a relatively high brightness needed. (we typically work in biological systems with hundreds of NV centers in conventional patient samples, anything that is less bright than several tens of NVs wouldn't be visible in a medical sample) Do I remember correctly that these defects are much less bright than NV centers? If that's true, I would say the claim that it is for biological and biomedical applications is overly optimistic. (that doesn't mean the material isn't promising for other applications) EDIT: below it says “typical saturation count rates in the range 5-200 kcps” so this is indeed very low when compared to NV centers.

Reply:

We have compared the brightness of single NV-centres in nanodiamonds and the carbon-related hBN defects used in this study on the same confocal microscope. The results are shown in Fig. R1, below. Here, the count rate at saturation (I_{sat}) for five NVs on our setup is between 75-100 kcps. For the carbon-related hBN defects, the I_{sat} is typically 5-200 kcps, and in the case shown 620 kcps. Thus, the hBN defects show brightness that is comparable to NV centres, in some cases dimmer and some cases brighter. We have changed our wording in the manuscript to represent the range of I_{sat} values we have measured (line 118 of full marked up version). We have also included Fig. R1 in the revised Supplementary Information (Fig. S3).

Regarding biological sensing, the reviewer makes the point that enough signal needs to be collected in biological media for sensing to be possible. For ensemble sensing, the sensitivity improves with defect number¹, n , by $\frac{1}{\sqrt{n}}$. This means sensing with 100 NVs gives rise to 10x improvement in sensitivity compared to one NV. For the carbon-related hBN defect, with the range of possible PL countrate and ODMR contrast (up to 90%) that we report, the DC sensitivity can approach a 10-fold improvement on that of a single NV centre. For example, the DC sensitivity for single defects is proportional to $\frac{1}{C \sqrt{PL}}$. If we compare an NV centre with PL = 90 kcps and C =30% to an hBN defect with PL= 500 kcps and C=90%, the hBN defect can represent a 10-fold improvement in sensitivity.

Considering the reviewer's question, we acknowledge wording used in the manuscript could give rise to confusion between ensemble/nanoscale sensing. We have therefore removed the reference to biological quantum sensing from the introduction (lines 53-54).

Figure R1: The saturation of photoluminescence (PL) intensity (I) (in kcps) for five single NV centres (black circles) and one hBN defect (blue circles). All data sets are fit to $I = \frac{I_{sat}P}{P + P_{sat}}$, where I is the PL intensity in kcps, I_{sat} is the countrate at saturation, P is the optical power and P_{sat} is the optical power at saturation, both in μW .

3. *“This is in stark contrast to what is seen for the NV centre, where a magnetic field ~ 10 mT misaligned to the defect’s quantization axis quenches the ODMR resonances due to degradation of the spin initialisation mechanism” this is true. But as it stands now the article is a bit overly critical with NV centers.* “

Reply:

Criticism of the NV system is not the message we were aiming for in our manuscript, so we thank the reviewer for alerting us to this. In our original manuscript, we intended to outline that NV magnetometry is an extremely powerful tool, which is, to-date, unrivalled by other defect systems for nanoscale quantum sensing. In our revised version we have made significant modifications to the introduction (lines 61-67) and included a new Figure 4 that no longer shows a direct comparison with the NV centre.

4. *“The overall sensitivity of NVs is still much higher and brightness is a real issue in boron nitride. There should be a more balanced discussion which includes the downsides of the boron nitride material.”*

We agree with the reviewer that the manuscript would benefit from a discussion of the current drawbacks of the hBN system. The main drawbacks are that the hBN system represents a less mature technology that still requires significant further optimisation and engineering before it will reach the level of reproducibility of the NV platform. The large variation in brightness and ODMR contrast on the single defect level (as presented in our manuscript) means reliable device fabrication is currently challenging. Future work requires developing methods of

deterministic defect creation and control, as has already been achieved in the diamond community. In addition, the relatively low spin coherence times compared to isotopically purified diamond will place the upper limits on the sensitivity for DC and AC sensing. We have included these points and modified the text to give more balanced discussion of the hBN system in the revised manuscript (lines 436-443).

Our response above has addressed the relative brightness of the two systems. Regarding sensitivity, the state-of-the-art sensitivity achieved with NV centres for DC magnetometry is $72 \text{ nT Hz}^{-1/2}$ ², which is for deeply embedded NV centres. For shallow (80 nm deep) NV centres, which are more comparable to the hBN defects (our materials are 30 nm thick), the best sensitivity achieved is $300 \text{ nT}\sqrt{\text{Hz}}$ ³ and typical DC sensitivities are $\sim 1 \mu\text{T Hz}^{-1/2}$ ⁴. In our revised manuscript we show experimentally that DC sensitivity for the hBN defects reaches $1.5 \mu\text{T Hz}^{-1/2}$. We have included new discussion of the experimental sensitivity at lines 341-342 of the revised manuscript, and in the Supplementary Figure S23.

Reviewer #2 (Remarks to the Author):

1. Gilardoni et al performed cw ODMR studies of carbon-related spin defects in hBN. They discover that ODMR resonances are still detectable even when the field is applied off-axis. They assign this to low symmetry of the defect's spin density. They further simulate achievable magnetic-field sensitivity using these defects.

Spin defects in hBN have attracted much attention recently due to their high ODMR contrast at room temperature. Applying them in quantum sensing seems like a logical next step. The manuscript is well written and the results are clearly presented.

Reply:

We thank the reviewer for their time in assessing our manuscript. We are encouraged that the reviewer finds the quality of the submitted manuscript high.

2. However, I have reservations towards its publication in Nature Communications. ODMR features of this type of defects have been reported by several groups, including some of the co-authors on this manuscript.

Reply:

The defects with the ODMR features reported here have only been reported in one other recent reference, by the same authors (Stern et al., Nat Materials 2024). However, hBN indeed contains more than one defect type that is being pursued for quantum sensing. These include the negatively boron vacancy (V_B^-), which has GHz-zero field splitting ($D= 3.5$ GHz), but is too dim to be used on the single defect level^{5,6}. The V_B^- is a promising widefield quantum sensor but is not able to perform nanoscale magnetometry. Separately, recent work has presented singly addressable hBN spin defects with a S-1/2-like spin structure⁷. These defects do not show appreciable zero-field splitting therefore are unable to detect the orientation of an external magnetic field and therefore cannot be used for vectorial magnetometry. Publication of this work occurred after our initial submission.

Discussion of the V_B^- was included in the original manuscript (lines 84-85 of revised manuscript). To make a clear distinction between the different single hBN defects used for sensing, we have included additional discussion of the S-1/2 defect to page 4, lines 89-92.

3. The quantum sensing discussion, though provides a new angle, were only performed using simulations rather than experimental demonstrations. Overall, I find the novelty/ potential impact of this work to be limited.

The main result of our manuscript is the identification of a new multi-axis magnetic field sensor that is capable of vectorial field sensing, with high dynamic range (> 100 mT) and comparable DC sensitivity to typically used nanoscale sensors ($1.5 \mu\text{T}/\text{Hz}^{1/2}$).

This is important because the ODMR of NV-centres and other uniaxial spin-defect systems are limited to off-axis B-fields of under 10 mT and require computational extraction to extract a vector field map. This creates new opportunities for vector-field sensing that are unavailable in other defect systems.

These conclusions are supported by the experimental data reported in revised Figs 1-3. In the final section, we discuss how to extract the vector field from the measurements based on the calibration data, and this is of a theoretical nature. We recognise that the first version of our manuscript placed more emphasis on the results of the kinetic model than the data we

presented. In the revised version of the manuscript, we present and discuss the experimentally determined sensitivity (lines 341-342).

4. I also list a few comments that the authors may consider addressing to further improve the quality/completeness of the manuscript:

- The authors attribute the weak field orientation dependence of the ODMR signals to low symmetry of spin densities. I find this argument arbitrary and lacking proof. The authors should consider connecting this observation to the atomic structure of the defects, and if possible, perform DFT calculations to support this major claim.

Reply:

We assume that ‘weak field orientation dependence’ refers to the observation that the ODMR contrast is not quenched with an off-axis field. In the manuscript, we assign the origin of this behaviour to the strong spin-dependence of the intersystem crossing rates that emerges from our kinetic model.

Lines 246-248 state, ‘*The rates extracted using the procedure outlined above for the hBN defects cluster in the top right of the 2D plot, showing that these defects are characterised by strong spin-selectivity in both direct and reverse intersystem crossing processes. As a result, spin mixing requires a larger applied magnetic field in order to disrupt the optical spin initialisation mechanism, giving rise to large magnetic field dynamic range for the hBN sensor.*’

We acknowledge that some of the wording in the manuscript could have led to confusion on this point. For example, ‘*As observed experimentally, in the zero-field limit, the system is initialised into the $|Gy\rangle$ state, a direct consequence of the low symmetry observed in this system*’ (line 298). The second part of this sentence has been removed. In addition, ‘*asymmetric intersystem crossing*’ (line 193) has been replaced with ‘*strongly spin dependence intersystem crossing*’, to reflect that while we believe the symmetry of the triplet states may be responsible for the relative ISC rate magnitudes (as seen in other S-1 systems), is it speculation at this stage.

However, the low symmetry of the hBN defect system is unambiguously related to the presence of three ODMR transitions at zero field. It is well known that the low symmetry of the spin density is directly related to the presence of a significant transverse zero field splitting term (E), which mixes the + and – eigenstates at zero field and permits a transition between them. This results in an orthogonal quantisation axis for the spin defect which has significant advantages for magnetometry (refs 13-15 of the main text).

The reviewer is right that DFT would enable further insight into the ISC rates, however without knowing the exact atomic structure we cannot run DFT at this stage. This points to important future work, which we are following with high interest and discussing with theorists in the community.

5. The global model provides a reasonable fit in fig. 2b, but the fits in fig. 2c and 2d are quite poor, suggesting additional energy levels or dynamics not included in the model. The authors should provide possible explanations for this.

Reply:

We have performed new analysis of the data in Figure 2 and modified the figure in response to this comment and the comments from Reviewer 3. To avoid repetition, please refer to the response to Reviewer 3 (questions 1 and 2) (below).

Reviewer #3 (Remarks to the Author):

1. The manuscript by Gilardoni et al investigates an ODMR-active spin defect in hBN previously discovered by the same group, focusing here on its photodynamics and ODMR contrast as a function of magnetic field orientation, showing experimentally that ODMR contrast is preserved for a wide range of orientations, and predicting that a single defect can allow vector magnetometry. While the results are interesting, I find the claims insufficiently supported by the analysis, and so I cannot recommend publication in Nat Comms at this stage.

My main concern is about the analysis of the photodynamics, which is incomplete and unconvincing in my opinion. The authors chose to apply a global fit to the three normalised curves presented in Fig. 2 in an attempt to validate the model and estimate the parameters, but the fit in Fig. 2d is clearly not good which puts the validity of the model and/or parameters into question. At the very least there should be an uncertainty analysis on the extracted parameters listed in table 1, to reflect the poor fitting in Fig. 2d and show which parameters are the most uncertain as a result.

Reply:

We thank the reviewer for taking the time to evaluate our manuscript. We respectfully disagree with the reviewer that our analysis of the photodynamics is incomplete, and we lay our argument out below. However, we agree that further information on the fitting procedure would improve the manuscript. As suggested by the reviewer, we have provided below analysis of the uncertainties associated with the extracted parameters, more information about the global fitting procedure, as well as the reasons for the model selected.

First, we would like to clarify that our fitting procedure is not limited to the three datasets shown in Figure 2. Our modelling extracts a series of rates from a global fit to:

1. cw ODMR contrast (this provides an experimental bound)
2. saturated PL countrate (this provides an experimental bound)
3. $g^{(2)}(\tau)$ (original figure 2a)
4. relaxation experiment (original figure 2d)
5. Initialisation experiment (original figure 2c)

In Figure R2 (below) we show the results of our global fit to the five datasets listed above. The raw data is the black circles, and the green curves are the fit that is the same as shown in the original manuscript. The extracted parameters from this fit and their associated error are shown in Table R1. As can be seen, the results of the $g^{(2)}(\tau)$ and initialisation experiments are well captured by the model. As reviewers 2 and 3 point out, for the spin relaxation experiment there is some deviation between the best fit and the data at the early time delays.

In Fig. R2, in addition to the best fit (green curves, ie the original fit in the main figures) we show curves simulated using the model and a random set of combinations of parameters that give an error within <3 x the best fit (orange and blue lines) if compared to the experimental values. The distribution of optical rates that arise from these curves are shown in Figure R3 (here we show the log of the rates, which are the parameters utilised in the model). These are colour and size coded. Larger, blue circles represent smaller error while smaller, yellow circles represent configurations where errors are larger, and the colour coding matches the colours in R2. The green vertical lines on Fig. R3 represent the optical rates of best fit, which are in Table 1. The mean of the distribution (shown in red in Fig. R3) for each parameter are also printed in Table 1. In some cases, the best-fit values of the parameters do not coincide with the peak of the probability density for the parameters. We extract the error values for each parameter from the regions containing 99% of the area under the probability density functions (indicated by the vertical grey lines in Fig. R3). We present these in Table 1, and these are the same errors

reported in the revised main text. The parameters that show the highest uncertainty is the reverse intersystem crossing rate (from singlet to GS), with low-error occurrences spanning several orders of magnitude and peak-value of the probability density occurring close to the best-fit value reported by us originally in the manuscript. This indicates that the absolute value of the total reverse ISC rate is not bound by our model, and this rate has little influence on the experimental data at hand. Despite this, the value of $k_{ISC,GS}$, which sets the spin asymmetry in the reverse ISC process, is properly bound by our model and limited to 0.26, indicating strong spin-asymmetry in the reverse ISC as initially predicted by us. In conclusion, given our set of experimental observables that we see as hard limits to this global fit, the parameters we present in the text is the best fit of the proposed model to the data.

As stated above, the prediction of the relaxation curve that is obtained from the best fit to our data shows some deviation from the relaxation data at early time delays (panel (b) in Figure R2). Certain combinations of rates do improve the fit of the model to the early spin-relaxation data; however, at the same time, these combinations incur in a poorer fit to the long-time delays in the $g^{(2)}(\tau)$ data. One possible explanation for this trade-off may be that our kinetic model is an oversimplification of the level structure for this defect. It is possible that other higher lying excited states, in either the singlet or triplet manifolds, are involved in the optical cycle and influence the photoluminescence traces we measure. To explore this possibility, we experimented with relaxing the constraint of the count rate in the model. The best fit and fit from the mean values are shown in Figures R4 and R5. We find that if we relax the constraint on the defect brightness, the model can fit both the $g^{(2)}(\tau)$, spin-relaxation and initialisation data. However, the overall brightness is not well modelled, and this combination of rates results in a system that is at least 2-fold dimmer than our experimental data show. In our original manuscript we considered other possible optical level structures that may be applicable to this system. In our original Supplementary Information, we presented a model for a singlet ground state and triplet metastable state system, but the model does not qualitatively capture the behaviour of the system, thus we concluded a triplet ground state model is most likely. Other possible level structures may include other radiative pathways, in either the triplet or singlet manifold, or charge transfer states, but many different combinations are possible, and each would introduce multiple new free parameters.

To determine the effect of the uncertainty in the obtained optical rates on the main messages of the paper, we calculated the magnitude of ODMR contrast vs B_x , B_y , B_z for the defect in Figure 2. The result of this analysis is shown in Figure R6. We find that all four subtly different combination of rates predict that the ODMR contrast across f_a , f_b , f_c transitions for this defect is not quenched with strong transverse magnetic field of up to 200 mT. This means that one of the main conclusions of our paper, i.e. that this defect preserves ODMR contrast even under strong off-axis fields, is predicted by our model even if we consider the uncertainties in determining the parameters. This finding by the modelling is in accordance with our experimental data.

In conclusion, we hope the reviewer agrees that the overall approach we have taken in extracting our estimate of the optical rates of this system is complete and justified. We find that the model we present in the main text is the best fit to a large amount of experimental data and models the field-dependence of the ODMR behaviour well. Any deviations of the best fit and experimental data may be explained by other optical states that affect the brightness but not the ODMR response. In our updated manuscript we have included further discussion of our fitting procedure, including the errors and uncertainties involved:

1. Lines 202-203 include more details on the measurements used in the global fit.
2. Supplementary Section V-F has been added
3. Figure 2(b,c,d) have been modified to represent the uncertainty to the fits.

Figure R2: (a) $g^{(2)}(\tau)$, (b) spin-relaxation and (c) initialisation results. Black circles are experimental data, green curves are calculated value using the model described in the main text and parameters obtained from a fitting procedure. Red curves are calculated using the mean of the distribution of parameters from probability densities presented in Fig. R3. The yellow to blue curves are calculated using combinations of parameters obtained by random sampling, and where the overall and experiment specific errors are smaller than the error of the best fit (green curves).

Figure R3: Probability density functions of each of the 9 parameters involved in the fitting procedure, obtained by investigating $>1 \times 10^6$ samples randomly obtained combinations of parameters, and calculating both overall and experiment-specific errors. Here, we histogram the parameter values that give both overall and experiment-specific errors below $3 \times$ the error of the best fit. The circles corresponding to each sample are colour and size coded: samples that give smaller errors are given by large blue circles, whereas samples that give larger error are given by small yellow circles. The green vertical lines represent the parameter values of the best-fit to the data. Red curves are guide to the eye representing a probability density function obtained from the histograms. Gray vertical lines indicate the regions containing 99% of the area under the curve of the pdf.

Figure R4: As in Fig. R2, but with parameters obtained by relaxing the constraint of calc. PL > exp. PL.

Figure R5: As in Fig. R3, but with parameters obtained by relaxing the constraint of calc. PL > exp. PL.
 $k_{1,ES} = (\Gamma_{Ex-S0} + \Gamma_{Ez-S0}) / \Gamma_{Ey-S0}$, $k_{1,GS} = (\Gamma_{S0-Gx} + \Gamma_{S0-Gz}) / \Gamma_{S0-Gy}$, $\Gamma_{ISC,ES} = \Gamma_{Ex-S0} + \Gamma_{Ex-S0} + \Gamma_{Ey-S0}$,
 $\Gamma_{ISC,GS} = \Gamma_{S0-Gx} + \Gamma_{S0-Gz} + \Gamma_{S0-Gy}$.

Figure R6: Simulation of the magnitude of the ODMR contrast vs. B_x , B_y , B_z from 0 to 200 mT for all three resonances (fa, fb and fc), considering rates BF1, M1, EL1, and EU1, as defined in Table 1. Colour coding is the same as in Figure 3 of main text.

Table R1

Rates (MHz)	Γ_{G-E} (kHz/uW)	Γ_{E-G}	Γ_{Ex-S0}	Γ_{Ey-S0}	Γ_{Ez-S0}	Γ_{S0-Gx}	Γ_{S0-Gy}	Γ_{S0-Gz}	Γ_{T1}
PL constrained									
Best fit (BF1)	0.9	163	5.4	190	5.4	0.0022	0.676	0.0022	0.0032
Mean (M1)	0.4	136	29	227	29	0.007	0.561	0.007	0.004
Error [EL1 EU1]	[0.17 0.98]	[79 243]	[1 125]	[127 376]	[1 125]	[0 1E5]	[0.146 1E6]	[0 1E5]	[0.0028 0.0076]
PL unconstrained									
Best fit (BF2)	0.18	139	32	241	32	0.0052	0.275	0.0052	0.0032
Mean (C2)	0.19	139	39	289	39	0.020	0.721	0.020	0.0048
Error	[0.1 0.67]	[79 251]	[0.4 180]	[158 488]	[0.4 180]	[0 1E5]	[32 1E6]	[0 1E5]	[0.003 0.007]

2. Additionally and importantly, I find it strange that the authors do not show and analyse the sig and ref traces that led to Fig. 2c and 2d. Just showing that the difference is well fit by the model (for Fig. 2c) is not enough, the authors also need to show the model explains the individual traces before normalisation. Those can be background subtracted if needed (by running the same sequence away from the defect under study). For Fig. 2c, the raw traces correspond to the PL during the laser pulse, which are dictated in part by the E->S0 rates.

Reply:

We have analysed the signal and reference traces individually and this data is shown in Figure R7 and R8. Here we show the experimental contrast, signal and reference data for the initialisation measurement (top two rows- the top row is a separate measurement with finer steps) and the relaxation measurement in row 3, without any background correction.

Figure R7: (First column) The difference, (second column) signal and (third column) reference traces for the initialisation experiment (top two rows) and the spin relaxation experiment. The blue circles show the experimental data, where the y-axis is shown on the right of the plot. The red lines are the best fit from the model and link to the y-axis on the left.

Both data sets were collected with a randomised sequence of time delays. This enables us to identify influences of drift or slow noise (longer than seconds) in the raw data. We did not take background measurements near the defect under study, as the background due to other defect

emission was low (< 2 kcps compared to the defect countrate). It is difficult to measure a background that can account for the drift and is distinct from the reference data. However, we can see in our analysis that drift was present in the spin relaxation experiment (shown in Figure R8), resulting in an underlying background to the signal and reference that is difficult to remove without modifying the data. This means the raw traces less meaningful to compare to the model in this case.

R8: Drift in the spin relaxation experiment (a) The signal and reference traces shown in measurement time order on the x-axis. The underlying background gives rise to a slow change in countrate during the sweep. (b) The signal and reference data ordered in time delay.

3. For Fig. 2d, the raw traces are in effect recovery curves, which should reveal the rates $S_0 \rightarrow G$. Given the authors estimate the rates $S_0 \rightarrow G_{x,z}$ and γ_{T1} to be a few kHz, there should be a slow recovery component on the corresponding time scale (~ 500 us) in the sig and ref traces underpinning Fig. 2d.

Reply:

The data in Fig. 2d does represent a recovery curve. There are multiple timescales involved in the recovery of the ground-state population, these are $S_0 \rightarrow G_y$ and $S_0 \rightarrow G_{z,x}$. However, because $S_0 \rightarrow G_{z,x}$ is ~ 3 orders of magnitude smaller (675 Hz vs 2 Hz) and has a small amplitude compared to the other timescale, it is difficult to detect experimentally. If these timescales and amplitudes were more comparable, we would resolve a biexponential decay in the recovery curve. Our fit to the data directly reflects this - it shows what we should measure given a system with the timescales we propose. In the revised manuscript we refer to the data in Fig. 2d as a spin relaxation experiment (line 203), to make it clearer that this data represents a recovery curve.

4. Without a proper analysis of the full data set and a discussion of the uncertainties in the model and rates, I don't think the conclusions of the paper are sufficiently justified, given the sensitivity predictions in Fig. 4 are based on the parameters inferred from Fig. 2.

Reply:

In light of the reviewer's comments, we have included a discussion of the uncertainties in the model and rates (as per the reply above), as well as additional analysis of the full data set in the new Supplementary (Section V-F). We hope that the reviewer agrees that this analysis is detailed and complete and that the conclusions of the paper are well supported by our modelling and data.

In addition, we have significantly modified Figure 4 in our revised manuscript. The new Figure 4 demonstrates how this defect would be used to perform vectorial sensing, whereas discussion of DC sensitivity is focussed on our experimental data (as per reply to reviewer 2, question 3)

5. Other comments:

- For Fig. 1e, you write, "We determine the field vector is at 51(1)deg from the defect z-axis, parallel to the yz plane, from field-dependent measurements." -> Please define the y axis. Does the yz plane coincide with the plane of the hBN layers? That should be explained. Also, in the inset of Fig. 1e, the schematic says 52deg rather than 51deg.

Reply:

Thank you for pointing the labelling error out, the schematic has been corrected. The y axis is in the plane of the hBN layers, as stated on line 173 of the manuscript. The x-axis is out of the plane of the hBN layers, and a statement has been included at line 174 to make this more clear. A modification to Figure 1 has also been made to highlight the orientation of the x axis.

6. In the main text you write "see Fig. S3 for zero-field contrast statistics for a range of defects" -> I think you meant Fig. S1

Reply:

Thank you. The error has been fixed.

7. In Fig. 3b,c the ODMR contrast is calculated as a function of magnetic field direction for a constant magnitude of 51 mT. Since a key point of the paper is the extended magnetic field range, I think there should be predictions as a function of the field magnitude, e.g. the ODMR contrast of each resonance could be plotted in a heat map as a function of (B_x, B_y) and another one as a function of (B_y, B_z) to extend Fig. 3b,c.

Reply:

Thank you for this suggestion, in the revised manuscript we have added subfigures b,d to the revised Figure 3 which show the ODMR heat maps as a function of B_x, B_y, B_z as the reviewer suggests. Further discussion of this figure is included in response to the reviewer's later question.

8. Likewise, the sensitivity predictions in Fig. 4 assumes a constant magnitude of 50 mT. I think the angular plots which are hard to read should be complemented by 2D heat maps as a function of (B_x, B_y) and (B_y, B_z) . This would also allow a more meaningful comparison with the NV case beyond just the special case of the ESLAC region.

Reply:

We appreciate the reviewer is concerned by a comparison of sensitivity between hBN and NV in the region of the NV ESLAC. As per reviewer's suggestion, we no longer present the angular plots with a comparison with the case of the NV, instead choosing to show the data in 2D colour maps (new Figure 3b,d).

We point out that there is nothing unique about the behaviour at 50 mT, indeed, for certain magnetic field orientations the ODMR contrast persists for fields of > 100 mT magnitude. The data in Figure 3a,c is presented at 50 mT because at this field we could experimentally access a wide range of field orientations.

9. It would also justify the claim in the abstract that “provide sub- $\mu\text{T}/\text{rt}(\text{Hz})$ magnetic-field sensitivity for both on and off-axis bias magnetic field exceeding 50 mT” (the word “exceeding” lacks justification).

Reply:

We understand the reviewer’s concerns. In the new Figure 3b,d we now present the ODMR contrast magnitude predicted for the hBN defect up to 200 mT.

10. For the sensitivity calculations, if I understand you took PL0 based on the rates inferred from Fig. 2 corrected by a factor 0.1 to account for collection losses. At optical saturation, this would mean a photon collection rate of $\Gamma_{E \rightarrow G} = 163 \text{ MHz}$ divided by roughly 2 to account for the ISC and multiplied by the 0.1 collection efficiency, which gives 8.5 Mcounts/s, far above the saturation count rates reported in the SI. There could be many reasons for this discrepancy (e.g. the $\Gamma_{E \rightarrow G}$ decay may have a non-radiative component), but in any case the authors should follow the common practice of using the measured photon rate (i.e. 200 kcounts/s) rather than an unverified prediction. This is especially important since you are making claims about sub- $\mu\text{T}/\text{rt}(\text{Hz})$ sensitivity in the abstract, which misleadingly suggests the defect outperforms NV.

Reply:

We entirely agree with the reviewer’s comments, and we assure them that in the original manuscript we did not use 8.5 Mcps in the calculations for sensitivity. In our calculations for sensitivity we used the predicted countrate (~ 270 kcps), not the radiative rate, and this gives rise to the sensitivity we present in the manuscript.

11. The explanation of how measurements with two different bias fields allow full vector magnetometry is not very clear to me. Fig. 4c plots sensitivity but that does not imply there is a unique solution for the vector magnetic field. The uniqueness of the solution needs to be discussed and a simulated set of ODMR spectra for a test vector field (plus two different bias fields) and its reconstruction would help greatly to illustrate the point.

Reply:

We appreciate that Fig. 4c was difficult to comprehend and we have made a revised Fig. 4 following the reviewer’s comments.

In this revised figure, we show, for a given 0.1 mT-target field (red vector), the ambiguity in its determination under two different 100-mT bias field configurations indicated by the pink and blue clouds in Fig. 4e. For an ODMR measurement with a bias field in the yz plane, possible target fields that could give rise to the ODMR spectrum are represented by the blue cylinder. For this single measurement, the B_y and B_z component of the target field can be accurately determined, whilst the B_x component cannot. A second measurement with a bias field in the xy plane, produces a range of possible target fields represented by the pink dome that restricts the possible ratios of the B_x , B_y and B_z components of the target field. Together, measurements in the two configurations give a unique solution to determining the target field.

Overall, this analysis shows that with this hBN spin, two bias field configurations can be used to determine unambiguously all three components of the target magnetic field. There are several key advantages this functionality provides for magnetometry:

1. Magnetic field maps can be measured without additional computational steps that are required to extract three B field components from a single axis sensor²⁰.

2. Measurement of the three components of the B field directly means that determination of magnetisation from the field map is less susceptible to noise and error^{8,9}.
3. Multiple complementary pairs of bias fields that can be used to perform this measurement, relaxing the requirement on applying bias fields in specific orientations with regards to the defect to perform sensing.

In summary, the hBN defect can detect all three components of magnetic field in a combination of conditions (ie. field magnitude, ambient conditions) that has provided limitations to other nanoscale sensors (defects and SQUIDs alike). With further engineering and development, the hBN defect will open new possibilities for nanoscale sensing across potentially a wide range of application area.

References:

- ¹ J. Taylor et al. Nat. Phys. **4**, 810–816 (2008).
- ² G. Balasubramanian et al., Nat. Mater. **8**, 383–387 (2009).
- ³ Q. C. Sun et al., Nat. Commun. **12**, 1989 (2021).
- ⁴ J. Rovny et al., Nat. Rev. Phys. **6**, 753–768 (2024).
- ⁵ A. Gottscholl et al., Nat. Mater., **19**, 540–545 (2020)
- ⁶ A. Gottscholl et al., Nat. Comm., **14**, 4480 (2021)
- ⁷ S. Scholten et al., Nat. Comms., **15**, 6727 (2024)
- ⁸ L. C. Bassett et al., Science, **345**, 1333–1337 (2014).
- ⁹ T.J. Smart et al., npj Comput. Mater., **7**, 59 (2021).